# Metformin Enhances TKI-Afatinib Cytotoxic Effect, Causing Downregulation of Glycolysis, Epithelial–Mesenchymal Transition, and EGFR-Signaling Pathway Activation in Lung Cancer Cells

**DOI:** 10.3390/ph15030381

**Published:** 2022-03-21

**Authors:** Pedro Barrios-Bernal, Norma Hernandez-Pedro, Mario Orozco-Morales, Rubí Viedma-Rodríguez, José Lucio-Lozada, Federico Avila-Moreno, Andrés F. Cardona, Rafael Rosell, Oscar Arrieta

**Affiliations:** 1Laboratorio de Medicina Personalizada, Thoracic Oncology Unit Instituto Nacional de Cancerología, S.S.A., San Fernando 22 Sección XVI, Tlalpan, Mexico City 14080, Mexico; darkpabb@hotmail.com (P.B.-B.); noryhp@yahoo.com (N.H.-P.); orozco81@hotmail.com (M.O.-M.); josemarialuciounam@gmail.com (J.L.-L.); 2Unidad de Morfología y Función, Facultad de Estudios Superiores (FES) Iztacala, Universidad Nacional Autónoma de México, Tlalnepantla, Mexico City 54090, Mexico; araceliviedma@hotmail.com; 3Lung Diseases and Cancer Epigenomics Laboratory, Biomedicine Research Unit (UBIMED), Facultad de Estudios Superiores (FES) Iztacala, Universidad Nacional Autónoma de México, Tlalnepantla, Mexico City 54090, Mexico; avilamore@hotmail.com; 4Foundation for Clinical and Applied Cancer Research—FICMAC/Molecular Oncology and Biology Systems Research Group (Fox-G), Universidad El Bosque, Bogotá 11001, Colombia; a_cardonaz@yahoo.com; 5Catalan Institute of Oncology, Germans Trias I Pujol Research Institute and Hospital Campus Can Ruti, 8908 Badalona, Spain; rrosell@iconcologia.net

**Keywords:** lung cancer, afatinib–metformin, EGFR, glycolysis, oxidative phosphorylation, epithelial–mesenchymal transition

## Abstract

The combination of metformin and TKIs for non-small cell lung cancer has been proposed as a strategy to overcome resistance of neoplastic cells induced by several molecular mechanisms. This study sought to investigate the effects of a second generation TKI afatinib, metformin, or their combination on three adenocarcinoma lung cancer cell lines with different EGFRmutation status. A549, H1975, and HCC827 cell lines were treated with afatinib, metformin, and their combination for 72 h. Afterwards, several parameters were assessed including cytotoxicity, interactions, apoptosis, and EGFR protein levels at the cell membrane and several glycolytic, oxidative phosphorylation (OXPHOS), and EMT expression markers. All cell lines showed additive to synergic interactions for the induction of cytotoxicity caused by the tested combination, as well as an improved pro-apoptotic effect. This effect was accompanied by downregulation of glycolytic, EMT markers, a significant decrease in glucose uptake, extracellular lactate, and a tendency towards increased OXPHOS subunits expression. Interestingly, we observed a better response to the combined therapy in lung cancer cell lines A549 and H1975, which normally have low affinity for TKI treatment. Findings from this study suggest a sensitization to afatinib therapy by metformin in TKI-resistant lung cancer cells, as well as a reduction in cellular glycolytic phenotype.

## 1. Introduction

Lung cancer (LC) remains the leading cause of cancer-related deaths worldwide, with an estimated 1.8 million deaths annually. Among the several subtypes of lung cancer, non-small cell lung cancer (NSCLC) is the most frequent and represents approximately 84% of cases. Several well-recognized risk factors have been associated with the incidence and molecular characteristics of lung cancer, such as tobacco smoking, wood smoke exposure, genetic susceptibility/ancestry, and air pollution, which altogether create heterogeneous genetic profiles of this malignant disease [1,2,3,4].

Consequently, NSCLC is characterized by several molecular alterations, including *EGFR* mutations, which are frequent (30–40%) in NSCLC, particularly in patients without a smoking history and wood smoke exposure [5]. Alterations in the *EGFR* gene can stimulate the intrinsic cytoplasmic kinase activity of this receptor, leading to constitutive activation of proliferative and survival signaling pathways, such as PI3K-AKT-mTOR, JAK-STAT, and RAS-MAPK, which are important for uncontrolled growth in LC cells [6,7,8]. This is the premise for the mechanism of action of several TKIs developed to overcome this overactivated signaling. All these targeted treatments are competitors for an ATP binding pocket at the kinase domain of the EGFR protein; this in turn decreases the phosphorylation of the receptor and attenuates the associated-pathway activation. However, several resistance mechanisms to TKI-treatment have been documented in clinical studies, such as secondary EGFR mutations (T790M and C797S), MET amplifications, epithelial–mesenchymal transition (EMT), and particularly the activation of the Warburg effect (WE), that promotes an important metabolic remodeling in tumor cells [9,10,11].

*EGFR*-mutated cancers promote WE, deriving in increased glucose uptake and lactic acid fermentation, which work indirectly as fuel suppliers in order to sustain cellular proliferation. In this sense, *EGFR* mutated (*EGFR*m) cells are characterized by high glycolytic activity with decreased oxidative phosphorylation; this metabolic phenotype is associated with increased metastases and cellular growth, as well as immune evasion. In addition, overactivation of EGFR signaling has been shown to promote EMT in LC cells, which is associated with resistance toTKIs [12,13,14,15]

The study of the aforementioned mechanisms represents a priority, considering that virtually all TKI-treated patients eventually develop resistance to this treatment modality. As such, several drugs have been proposed to overcome this resistance. In this context, metformin, a biguanide normally used to treat patients with type 2 diabetes, has been proposed as an anti-neoplastic agent, in combination with first generation TKIs for treatment of EGFRm NSCLC [16,17]

In detail, metformin stimulates the AMP-activated protein kinase (AMPK) indirectly, activating its catalytic subunit alpha 1, inhibiting mTORC1, decreasing translational protein synthesis, and blocking tumor cell proliferation [18,19,20]. Furthermore, some reports have demonstrated that metformin synergizes with first generation TKIs (gefitinib or erlotinib), reducing tumor cell growth and PI3K/AKT/mTOR axis activity in NSCLC cell lines with wild-type Liver Kinase B1 (LKB1) [20]. Evidence has shown that the combined treatment of metformin plus TKI-gefitinib promotes EMT-phenotype regression in TKI-resistant LC cells, increasing E-cadherin protein levels and blocking Interleukin-6 (IL-6) signaling [21]. Similarly, combined first generation TKI-metformin treatment also inhibits activation of both insulin-like growth factor (IGFR) and protein kinase B (AKT), showing a synergistic effect promoting apoptosis induction [22].

Combination of second generation TKIs with metformin has importantly improved the objective response rates of NSCLC patients, however, the mechanisms behind this association are poorly understood; therefore, this study aids to evaluate for the first time, the effect of the treatment with second generation TKI afatinib and metformin, as single agents, or in combination in LC cell lines. In order to reach this objective, we describe the molecular consequences of both drugs in terms of cytotoxicity, EGFR pathway, glycolytic phenotype, and EMT markers, along with the type of pharmacodynamic interaction between metformin and afatinib.

## 2. Results

### 2.1. Unique or Combined (Afatinib and Metformin) Treatments Decrease Cellular Viability of NSCLC Cell Lines in an Additive–Synergistic Manner

To determine the cytotoxic effects of afatinib and/or metformin treatments, we performed MTT cellular assays. We found that metformin can sensitize TKI-resistant LC cells H1975, which showed a trend towards increased cytotoxic induction by TKI-afatinib. Sensitive LC cells HCC827 showed cellular inhibition induced by combo number 1 (afatinib 3 nM plus metformin 8 mM) compared with TKI-afatinib (3 nM) treatment (38% vs. 60%, *p* = 0.0046). Interestingly, EGFR wild-type LC cells, A549, showed an increased cytotoxic effect using a combined TKI-afatinib plus metformin treatment, displaying a higher cytotoxicity using combination treatment number 3 (afatinib 8 µM plus metformin 15 mM) compared with TKI-afatinib alone (8 µM) (24% vs. 50%, *p* = 0.0054) (Figure 1A).

The pharmacodynamics of the drugs were then analyzed using Compusyn software 1.0, with the Chou–Talalay index (see Methods section). Compusyn plots indicated that three LC cell lines showed synergic and/or additive effects with their respective combination schemes using TKI-afatinib plus metformin (Figure 1B).

According to the scale proposed by Chou and Talalay, the combination index revealed that LC cells H1975 show synergic effects using 2 and 3 µM of afatinib plus metformin. Lung cancer cells HCC827 showed a slight synergistic mechanism with 3 nM, and finally, lung cancer cells A549 had synergism in all three combinations. None of the combinations showed antagonism in any LC cell line (Appendix A).

### 2.2. Combined Treatment of Afatinib Plus Metformin Induces Apoptosis in a Synergic Way

To corroborate the induction of cytotoxicity and cellular death mechanism, we performed apoptosis assays. All tested LC cells revealed an increase in apoptosis induction when metformin is added to TKI-afatinib therapy. LC cells H1975 exhibited higher apoptosis induction using combo number 1 compared with afatinib alone 1 µM (48% vs. 26%, *p* = 0.0001). LC cells, HCC827, showed the highest apoptosis rate induced by combo number 2 compared with TKI-afatinib alone, 4 nM (58% vs. 45%, *p* = 0.0041). Finally, LC cells A549 showed higher apoptosis in combo number 1 compared with TKI-afatinib alone, 6 µM (50% vs. 28%, *p* = 0.013) (Figure 2).

From this point, it is important to mention that we only evaluated the highest synergic combinations identified for each LC cell line, to test the molecular markers, glucose uptake, and lactate secretion (Appendix A).

### 2.3. EGFR Expression at the Cellular Membrane Is Regulated by Combined Afatinib–Metformin Treatment

We evaluated changes in the percentage of EGFR protein expression at the cell membrane by flow cytometry. Single staining was performed without compromising the cell membrane to identify the highest amount of protein in H1975, HCC827, and A549 cell lines. Lung cancer cells H1975 showed an EGFR protein reduction of 14% when treated with the combination therapy compared with afatinib alone (*p* = 0.0120). In contrast, LC cells HCC827 did not show differences in EGFR membrane expression when treated with the combination therapy or afatinib alone, since both treatments caused a significant reduction. Lung cancer cells A549 exhibited EGFR reduction when treated with the combination therapy, compared with TKI-afatinib alone, with a reduction of 15.7% (*p* = 0.0433) (Figure 3).

### 2.4. Metformin Might Sensitize LC Cells to EGFR-TKI Therapy through the Reduction in the Expression and Activation of the EGFR Signaling Pathway

To elucidate the impact of afatinib–metformin treatment on the EGFR signaling pathway, first we performed Western blot assays. We detected an inhibition of the EGFR signaling pathway activation in all metformin-treated NSCLC cell lines. H1975 cells showed inhibition in the total EGFR protein level, as well as for the phosphorylated levels of EGFR, AKT, and P70S6K (Figure 4). As expected, HCC827 cells showed an axis-inhibition by afatinib treatment due to its sensitivity, but interestingly, metformin also showed inhibition of the activated EGFR-signaling pathway proteins (Figure 4), as well as total proteins (Appendix A). While in A549 cells we observed a trend towards an increase in phosphorylated forms, when afatinib was administered alone, but, when combined with metformin, we observed a reduction in total protein levels, with the exception of AKT (Appendix A), and in the same way, the pathway-phosphorylated forms were inhibited (Figure 4).

### 2.5. Combined Metformin/EGFR-TKI Treatment Reduces EMT Biomarkers and Increases Epithelial Marker E-Cadherin in NSCLC Cells

Once we confirmed that inhibition of the EGFR signaling pathway was induced by the treatment combination with metformin–afatinib, we evaluated if the combination could have an impact on the expression of proteins related to the EMT phenotype and increase the epithelial marker e-cadherin. To this end, we carried out Western blot and zymogram assays to measure levels of key proteins and matrix metalloproteases (MMPs), respectively. H1975 cells showed inhibition for all EMT phenotype markers, mainly β-catenin (*p* = 0.0067 vs. control), using the combined treatment. HCC827 cells showed inhibition mostly in n-cadherin when exposed to the combination treatment (*p* = 0.0028 vs. control). Similarly, the combination treatment inhibited vimentin and n-cadherin in A549 lung cancer cells. Additionally, we observed an increase in e-cadherin with our treatments, mainly with metformin alone in the A549 cell line, and the combination in H1975 and HCC827 cell lines. Regarding MMPs protein expression, all LC cell lines showed a similar reduction in the inhibition pattern when treated with metformin or afatinib, but importantly, also when treated with the combination scheme (Figure 5).

### 2.6. The Combination Treatment with Metformin–Afatinib Modifies the Glycolytic Phenotype in an EGFR-Mutation Status Dependent Manner

To evaluate if EGFR-signaling axis downregulation induced by afatinib–metformin combination might modify glycolytic activity, we measured the expression of glucose transporters (GLUTs), Hexokinases (HXKs), AMPK, phosphorylated-AMPK, pyruvate kinase M2 (PKM2) enzyme, and OXPHOS subunits which are functionally key elements to define a glycolytic phenotype, and involved in cell glucose uptake and lactate secretion, of LC cells.

Our results showed a reduction in GLUT1 and GLUT3 expression using combination or single-agent treatments on H1975 cells. Combined therapy inhibited GLUT3 (*p* = 0.0003 vs. control) and inhibited GLUT1 on HCC827 cells. Similarly, A549 cells presented a reduction in both glucose transporters using metformin treatment, though the combined treatment had a greater reduction in GLUT1 and GLUT3 (*p* = 0.0348; *p* = 0.0011 vs. control, respectively) (Figure 6). None of the NSCLC cell lines showed changes in total AMPK levels (Appendix A). However, we detected a remarkable increase in *p*-AMPK levels when using the combination treatment (*p* = 0.0357 and *p* = <0.0001 vs. control, respectively) in A549 and H1975 cell lines. However, HXK1 levels did not show changes for all NSCLC cell lines; A549 cells did not show HXK2 basal expression levels, however, H1975 and HCC827 cells showed notable reductions in levels of HXK2 by metformin addition. Finally, A549 and HCC827 cells showed reduced levels of p-PKM2 by both metformin and combination treatments (Figure 6). Conversely, H1975 cells only had a p-PKM2 reduction with the combined treatment (Appendix A).

In addition, we explored the impact of metformin–afatinib treatment on glucose uptake by measuring cellular incorporation of 2DG6P and lactate secretion through its measurement in cell culture medium. LC cells H1975 had a reduction of 33.4 pmol with combined treatment compared with controls (*p* = 0.0009). LC cells A549 exhibited a decrease of 42.6 pmol for 2DG6P uptake with combination vs. control (*p* = <0.0001). Finally, LC cells HCC827 showed glucose uptake inhibition in a range of 20.9–29.9 pmol when treated with metformin alone and with the combined scheme, compared with control (*p* = <0.0001) (Figure 7A).

We did not identify a statistically significant difference in lactate secretion, as shown in Figure 7B, in H1975 LC cells; however, LC cells HCC827 and A549 displayed a significant reduction in lactate secretion of 6.0 ng/µL (*p* = 0.0005) and 4.6 ng/µL (*p* = 0.04), respectively, when exposed to the combination treatment compared with controls (Figure 7B).

Finally, we identified that these metabolic modifications are associated with a trend to increased levels of five subunits of mitochondrial complexes participating in OXPHOS, with a slight upregulation for NDUFB8 (complex I), MTCO1 (complex IV), and ATP5A (complex V) in all LC cells lines studied here (Appendix A).

## 3. Discussion

We sought to better delineate the effects of the combination of metformin with afatinib in *EGFR*-mutated NSCLC cell lines; to carry out this objective, our investigation identified a synergic cytotoxicity and apoptosis in all the different lung cancer cell lines after using afatinib–metformin combination, as metformin effects on non-transformed cells include inhibition of liver gluconeogenesis, increase and acceleration of glucose uptake, and stimulation of insulin receptors (IRs) [23,24,25].

Therefore, clinical evidence has shown better objective response rates by the addition of metformin compared to standard first or second generation TKI therapy [26,27].

Additionally, recent evidence has shown the clinical benefit of adding metformin to treatment with different generations of TKIs, thus prolonging PFS and OS in LC patients with or without diabetes mellitus [28,29]. The concurrent use of metformin could be useful in various clinical settings in which patients will exclusively have access to first and second generation TKIs due to the expensive cost of third generation inhibitors. Thereby, continued search for novel cost-effective therapeutic strategies represents a priority to improve outcomes in LC patients.

To date, the use of metformin is controversial for routine clinical therapy of patients with *EGFR*m NSCLC, due to heterogeneity and variability in terms of results from previous trials mentioned above. Additionally, the exact mechanisms by which metformin could exert antineoplastic activity in lung cancer cells has not been yet asserted.

In this regard, multiple reports have evaluated the effect of combining metformin plus first generation TKIs on LC cell lines, and results have demonstrated induction of cytotoxicity and improvement in overcoming resistance to EGFR-TKI gefitinib [11,22,30]. In line with this observation, our research has identified synergic cytotoxic and apoptotic effects in all tested LC cell lines treated with the afatinib–metformin combination. Interestingly, the low affinity TKI lung cancer cell lines, A549 and H1975, showed the highest synergic effects. In agreement, a study showed that the combination of erlotinib plus metformin inhibited cell proliferation and reduced xenograft tumor growth derived from an A549 cell line [31]. Nonetheless, a previous investigation measured the combination index of first generation TKI gefitinib plus metformin and found antagonistic action using A549 cells, with a synergistic effect for H1975 cells. Such variability in treatment seems to be dependent on TKI generation and its concentrations along with metformin. Interestingly, it has been described that high doses of drugs can modify the interaction between both treatments [9,16,20,31]. Regarding apoptosis analysis, we observed a death induction ranging from 10% to 60% in all treatments, which strongly correlates with the synergistic effect observed in our cytotoxicity assays. Furthermore, these findings also agree with previous studies showing apoptosis as the primary death-induced mechanism by afatinib, or metformin, both as monotherapies [32,33,34].

Some of the molecular mechanisms explaining our results on cellular survival include an important decrease in the activation of EGFR in H1975 and A549 cells, caused by the combination of metformin and afatinib treatment; however, we did not identify changes in total EGFR protein expression in A549 cells after treating them with afatinib alone, as this particular cell line does not present mutations in the EGFR gene, and previous studies have shown that higher concentrations of TKIs are necessary to inhibit p-EGFR levels, and its associated signaling intermediaries, in many EGFR-wild type and TKI-resistant lung cancer cell lines [35]. Regarding HCC827 cells, as expected, afatinib importantly reduced their phosphorylated proteins, as they are highly sensitive to this treatment, but more interestingly, metformin exhibited a higher inhibition of the EGFR axis than afatinib in this cell line. Therefore, we observed that the sensitization provided by metformin can decrease these TKI concentrations in resistant cell lines [35]. Additionally, we established an important relationship between the EGFR pathway with glucose metabolism and EMT. We observed that inhibition provided by afatinib–metformin treatment affects the activated forms of AKT, promoting downregulation of the glycolytic phenotype, and serine/threonine kinase P70S6K activity, which is an important downstream effector of mammalian target of rapamycin (mTOR). This phenomenon may be explained by the metformin-mediated activation of AMPK (dependently or independently of LKB1 expression), which attenuates anabolic processes, such as protein synthesis, through inhibition of ribosomal subunit P70S6K, thereby impairing the production of important proteins for tumor progression [12,36,37].

Reduction in levels of EMT and reestablishment of epithelial markers has been proposed as one of the regulated mechanisms by the use of metformin and its combination with first generation TKIs. Nevertheless, information regarding EMT phenotype downregulation in metformin treatments involving afatinib (second generation TKI) was still lacking; hence, we analyzed the expression of multiple EMT-related molecules to compare these results with those obtained by first generation TKIs. In this sense, we found a decrease in N-cadherin, vimentin, β-catenin, MMP-2, and MMP-9 levels by use of metformin alone or in combination with afatinib, along with an important increase in E-cadherin in response to all treatments, and mainly to this combination. According to each cell line, afatinib inhibited N-cadherin and β-catenin on H1975 cells, and only N-cadherin in HCC827 cells. These effects were not reachable for first generation TKI in previous studies, possibly due to unstable EGFR pathway inhibition and TKI resistance through activation of the IL-6R/JAK1/STAT3 axis [21,38,39,40]. In agreement, a significant reduction in EMT markers has been reported in TKI-resistant LC cell lines treated with the combination of metformin plus gefitinib [21,40].

There is evidence for crosstalk between the EGFR signaling pathway and the promotion of the Warburg phenotype; for example, a study showed that the promotion of aerobic glycolysis by the phosphatidylinositol 3-kinase (PI3K)/mTOR axis is strongly influenced by EGFR. To explore this association, they inhibited EGFR, which reduced lactate production, some molecular markers, and glycolysis-related metabolites [41]. These findings were corroborated by other studies using EGFR inhibitors in lung cancer cells, in which they observed a clear EGFR-dependent glycolytic metabolism phenotype adoption, as they observed important decreases in the expression of GLUT3 and HXKII proteins in response to treatment with first generation TKIs [12,42].

Similarly, metformin exerts multiple metabolic modifications, as it inhibits mitochondrial complex I of the electron transport chain, thereby increasing AMP concentrations, which activate AMPK, and finally inhibit P70S6K activation, thus stopping protein synthesis of multiple metabolic enzymes and transporters [43,44]. This phenomenon has previously been reported in the literature, as one NSCLC study suggests increased glucose consumption and cancer cell degradation in early stages [45]. On the contrary, there is some evidence that suggests inhibition of glucose uptake, resulting in a reduction in glycolytic processes and the regression to energy generation through OXPHOS processes [46,47,48].

Our research found thatcombination with metformin and afatinib, reduced glycolytic markers, glucose uptake, and lactate secretion for all lung cancer cell lines included here. Interestingly, A549 cells showed strong inhibition of GLUTs levels and activation of AMPK, independently of LKB1 expression. Studies have reported that multikinase inhibitors have the capacity to activate AMPK through sensing a dysfunction in the mitochondrial metabolism, also by the activation of alternative pathways such as Jun N-Terminal Kinase (JNK), independently of LKB1 expression [49,50].

Furthermore, metformin-mediated activation of AMPK inhibits protein synthesis induced by mTOR, explaining the reduction in total and activated forms of P70S6K and other proteins [51]. In agreement, a previous study exhibited that metabolism modifications might activate AMPK, resulting in inhibition of LC cells proliferation, independently of LKB1 expression [37]. On the other hand, our results showed that the H1975 cell line, without alterations in LKB1 protein, showed an important AMPK activation by metformin associated with a reduction in glucose uptake, lactate secretion, and GLUTs and HXK2 levels; this phenomenon has also been reported in other studies, when metformin is combined with other compounds, modifying the AMPK activation and downregulating the glycolytic pathway [52,53].

Conversely, HCC827 cells did not show AMPK overactivation by any treatment, probably due to its high basal expression; however, a study showed that acquired resistance to TKIs treatment in this cell line can reduce p-AMPK expression. Clinical relevance of the AMPK pathway has been documented in solid tumors, in which the lost and low expression of p-AMPK has been reported as a factor for worse prognosis and disease progression [30,54,55].

HCC827 cells showed significant inhibition of GLUTs, HXK2, and particularly, levels of P-PKM2 were reduced with metformin treatment. Such kinase plays a crucial role in the conversion of phosphoenolpyruvate (PEP) to pyruvate. PKM2 dimerization has the capacity to regulate the shift from normal to lactate-based respiration in tumor cells, making this protein a potential therapeutic target. Glucose uptake and lactate production were reduced in HCC827 cells by all treatments studied. This effect can be explained by the sensitivity of this cell line, HCC827, to TKIs-based treatment [56,57].

Finally, EGFR inhibition was reported to reverse from Warburg phenotype to OXPHOS metabolism. Some reports suggest that inhibition of the EGFR pathway can reestablish oxidative metabolism associated with a decrease in glycolytic markers as well as lactate production, limiting fuel for tumor cell proliferation [12,58]. Conversely, in clinical studies, OXPHOS acts as a promotor of TKI treatment resistance, possibly due to metabolic reprogramming caused by inhibiting EGFR [59,60].

In this respect, we explored the influence of our treatments on expression levels of subunits from the five complexes that participate in OXPHOS. We did not find any significant change in all cell lines; however, we note an increasing trend in some subunit levels. Previous evidence showed that OXPHOS upregulation through inhibition of the EGFR pathway is associated with an increase in intracellular ATP, and a decrease in PKM2 phosphorylation, stimulation of catabolism, and these mechanisms have an impact on tumor cell proliferation and protein synthesis [12].

Previous clinical studies reported that the combination of metformin and EGFR-TKIs increase TKIs response and survival rates in NSCLC patients. Complementary, our results suggest that metformin sensitizes lung cancer cells with low affinity for TKIs-based treatment in a synergic way. Although the mechanisms behind this type of pharmacodynamic interaction are not yet entirely elucidated, our research showed some important mechanisms for this association (Figure 8) and supports the effect observed in clinical studies, analyzing the merged effect of metformin and TKIs, allowing further investigation in effective treatment combinations [26,27,61,62].

## 4. Materials and Methods

### 4.1. Cell Lines and Treatments

All NSCLC cell lines were purchased from ATCC (Virginia, Manassas, USA), which contain different EGFR mutational statuses, conferring a differential pattern of sensitivity to EGFR-TKIs treatment. The NSCLC cell line A549 possesses wild-type EGFR, while H1975 cells have a double mutation of EGFR (T790M exon 20 and L858R exon 21), as well as HCC827 cells include an EGFR deletion in exon 19. NSCLC A549 cells were grown in F12 medium (Gibco, Waltham, Massachusetts. USA. 21700-075). Meanwhile, H1975 and HCC827 cells were maintained in supplemented RPMI 1640 medium (Gibco, Waltham, Massachusetts. USA 31800-022). Both culture media were supplemented with 10% fetal bovine serum (FBS) (Gibco, 26140-079) and 1% of penicillin–streptomycin–amphotericin B antibiotics (MP Biomedicals. Fountain Pkwy, Ohio. USA. 091674049). All lung cancer cells were incubated at 37 °C in a humidified atmosphere with 5% CO_2_.

NSCLC cell lines were treated with afatinib (LC laboratories. Woborn, Massachussets. USAA-8644) and metformin (Sigma Aldrich. St. Louis, MO, USA. PHR1084) for 72 h, according to different afatinib concentrations combined with metformin according to each NSCLC cell line, as shown in Table 1.

### 4.2. MTT Assay

To evaluate the cytotoxic effects of metformin–afatinib treatments, we performed MTT assays using thiazolyl blue tetrazolium bromide (Sigma Aldrich. St. Louis, Mo, USA.M2128). For this purpose, A549, H1975, and HCC827 NSCLC cell lines were seeded by triplicate for each treatment with a cellular density of 1 × 10^4^ in 96-well plates, and they were incubated in medium supplemented with 10% FBS at 37 °C in an atmosphere with 5% CO_2_. Once cells adhered, they were treated according to the previously mentioned concentrations shown in Table 1.

After 72 h of treatment, the medium was aspirated from wells, and it was replaced by 10 µL of MTT solution (prepared at 5 mg/mL) and 90 µL of medium (complemented with 10% of FBS). Then, the plates were incubated at 37 °C for 4 h. After, the medium with MTT was aspirated from the wells, and the formazan crystals were solubilized, 200 µL of DMSO-Isopropanol solution (1:1) were added to the plates. Finally, formazan absorbances were measured by a microplate reader at 570 nm (BioTek, Saint Clare, CA, USA, ELX 808).

### 4.3. Drug Interaction Analysis

To determine if the combined therapy of afatinib and metformin produces synergistic effects on cytotoxicity of LC cell lines, we used the software Compusyn 1.0 (Biosoft, Cambridge, UK) to obtain the combination index (CI) corresponding to each drug interaction, based on the method proposed by Chou and Talalay. Quantitative values of a CI less than 1 reveal a synergic effect. In contrast, if the CI ranges from 1 to 1.10, the combination possesses a probable additive mechanism (antagonism mechanism: when the treatments are quantitatively located above the threshold line; additive mechanism: when it is located close to the threshold line, and synergic effect when the treatment points are under the threshold line). Finally, CIs greater than 1.10 quantitatively suggest antagonism.

### 4.4. Apoptosis Assay

To determine the treatment-induced apoptosis in NSCLC cells, they were seeded in 24-well plates using a confluence of 5 × 10^4^. Once attached to the plate, they were treated according to the concentrations described (Table 1) and incubated for 72 h at 37 °C and 5% CO_2_. Afterwards, cells were detached using trypsin, washed twice with 1X Phosphate Buffer Saline solution (PBS), and later were marked to phosphatidyl-serine and propidium iodide by Apoptosis Kit “Annexin-V-FLUOS Staining Kit” (Roche, Basel, Switzerland, 11988549001) through flow cytometry (Life Technologies, Waltham, MA, USA) following instructions by the manufacturer.

### 4.5. EGFR-Membrane Quantification Assays

EGFR expression on the cell membrane was detected by flow cytometry. Briefly, cells were seeded at a confluence of 5 × 10^4^ in 24-well plates, treated according to combined treatments (Table 1), and incubated for 72 h. Then, cells were trypsinized, washed twice with 1X PBS, and fixed with cytofix buffer solution (BD 554655). Subsequently, cells were washed once using PBS solution and stained with EGFR-targeted antibody (BD 555997) for 30 min; after that, the excess of antibodies was washed once with PBS, and detection of EGFR-attached antibodies was performed by flow cytometry method.

### 4.6. Western Blotting

Cell lines were seeded at a confluence of 3 × 10^5^ in 6-well plates, then treated with the previously mentioned concentrations (Table 1) and incubated for 72 h. Later, cell lines were lysed using the RIPA lysis buffer system (Santa Cruz Biotechnology. Dallas, TX, USA. SC-24948A), according to the manufacturer’s instructions. Cell lysates were used to quantify proteins by Bradford (Bio-Rad, Hercules, CA, USA, #5000205) assay. Equal amounts of proteins were loaded in polyacrylamide gel wells and analyzed by the SDS-PAGE system (25 ng for total and 50 ng for phosphorylated proteins). Subsequently, lysates were transferred to pre-cut 0.2 µm nitrocellulose membranes using a Trans-Blot Turbo System (Bio-Rad, Hercules, CA, USA). These membranes were then incubated overnight with specific primary antibodies (Santa Cruz Biotechnology, Dallas, TX, USA) against EGFR (SC-311550; 1:1000), P-EGFR (SC-101669; 1:1000), AKT (SC-5298; 1:1000), P-AKT (SC-514032; 1:1000), GLUT-1 (SC-3777228; 1:1000), GLUT-3 (SC-74399; 1:1000), HKI-I (SC-46695; 1:1000), HKI-II (SC-374091; 1:1000), GAPDH (SC-47724; 1:10,000), and, E-cadherin (SC-8426; 1:1000). Additionally, specific primary antibodies (Cell signaling. Danvers, MA, USA.) against P70S6K (9202S; 1:1000), P-P70S6K (9205S; 1:1000), Vimentin (5741S; 1:1000), N-cadherin (13116S; 1:1000), β-catenin (8480S; 1:1000), AMPK (2603S; 1:1000), P-AMPK (25315S; 1:1000), P-PKM2 (3827S; 1:1000), and OXPHOS cocktail (Abcam, Cambridge, UK, #110413; 1:1000). The next day, nitrocellulose membranes were washed using PBS to remove excess antibodies, and then 5% BSA solution containing specific secondary antibodies (Li-Cor, Lincoln, NE, USA. 1:5000) was added. After one-hour incubation, membranes were washed three times to remove excess antibodies and analyzed with the blotting scanner C-Digit (Li-Cor, Lincoln, NE, USA). The bands were then quantified with ImageJ software (NIH).

### 4.7. Zymogram Assays

Cell lines were treated and incubated in 6-well plates for 72 h. MMP-2 and MMP-9 were analyzed by gelatin zymography. Cells were homogenized in cold lysis buffer with protease inhibitors. The supernatant was separated by electrophoresis using 10% SDS-polyacrylamide gels containing 0.15% *w*/*v* of gelatin, under non-reducing conditions. After electrophoresis, the gels were washed twice with 2.5% Triton X-100 for 30 min at room temperature to remove the sodium dodecyl sulfate (SDS). Gels were then incubated at 37 °C for 24 h in an activator buffer. MMP activity appeared as a clear band contrasting with a blue background. Antibodies for MMP-2 (Cell signaling 87809S; 1:1000) and MMP-9 (Cell signaling. Danvers, MA, USA. 13667S 1:1000) were detected.

### 4.8. Glucose Uptake Assay

Cells were seeded in 6-well plates at a confluence of 1 × 10^5^ and incubated for 24 h. Afterwards, the initial culture medium in the plates was replaced with 2% SFB-supplemented medium to starve the cells, and the plates were incubated overnight. The following day, the mentioned medium was replaced with Krebs Ringer Phosphate Hepes (KRPH) buffer, along with metformin, afatinib, or its combination, and the plates were incubated for 3 h. Afterwards, 2-deoxy-d-glucose-6-phosphate (2DG6P) without insulin was added to the wells, and the plates were incubated for 35 min. Subsequently, the kit (Sigma Aldrich. St. Louis, MI, USA, MAK083) protocol was continued, according to instructions by the manufacturer.

### 4.9. Lactate Secretion Assay

Cells were seeded at a confluence of 1 × 10^5^ in 6-well plates and incubated for 24 h. Next, lung cancer cells were treated, incubated for 3 h, and processed as was described by the kit protocol manufacturer instructions (Sigma Aldrich. St. Louis, MI, USA, MAK064).

### 4.10. Statistical Analysis

Data are shown as mean ± standard deviation (SD). Data provided by the MTT and Annexin V/propidium iodide assays were analyzed through two-way analysis of variance (ANOVA). Results from proteins, glucose uptake, and lactate secretion assays were analyzed using one-way ANOVA tests. All mentioned analyses were performed with Tukey’s post-hoc tests using the GraphPad software 7.04 (Scientific, San Diego, CA, USA). A *p* value ≤ 0.05 was considered statistically significant.

## 5. Conclusions

This study demonstrates that the combined treatment of metformin plus afatinib sensitizes TKI-resistant lung cancer cells to afatinib action; thus, lowering the inhibitory concentrations of this TKI. Furthermore, this combination increased cytotoxicity and apoptosis in a synergistic way. Similarly, HCC827 cells, commonly sensitive to TKIs, exhibited enhanced cytotoxic effects induced by using metformin with afatinib. Other important effects of this merged treatment are inhibition of EGFR pathway intermediaries (PI3K, AKT, and P70S6K) and decrease in glycolytic markers, along with a trend to increase OXPHOS proteins. Complementarily, afatinib promoted inhibition of mesenchymal markers, and increased the levels of E-cadherin, as single therapy, and in combination with metformin, thereby corroborating previous findings with first generation TKIs. All above findings expand the understanding of the relationship between EGFR signaling, EMT, and metabolic phenotypes involved in TKI-resistance mechanisms.

## Figures and Tables

**Figure 1 pharmaceuticals-15-00381-f001:**
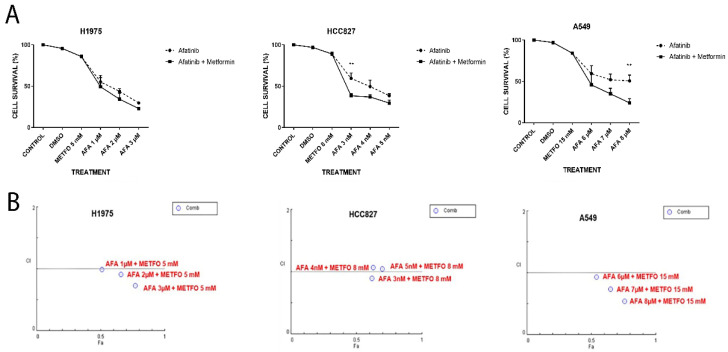
(**A**) Cytotoxic effect of afatinib alone or in combination with metformin in H1975, HCC827, and A549 NSCLC cell lines. Cells were seeded and treated with the previously described schemes for 72 h and MTT assays were performed. Points represent the mean of 3 independent experiments by triplicate. Statistical analysis was performed through two-way ANOVA. ** *p* ≤ 0.01. (**B**) Combination index plots from NSCLC cell lines. Plots show the different afatinib concentrations for each cell line, in combination with metformin. We observed that the H1975 cell line had synergism with the two highest concentrations of afatinib (2 and 3 µM), the HCC827 cell line had a degree of synergy with the lowest afatinib concentration (3 nM), while a synergic effect in the three combined treatments of the A549 cell line was observed.

**Figure 2 pharmaceuticals-15-00381-f002:**
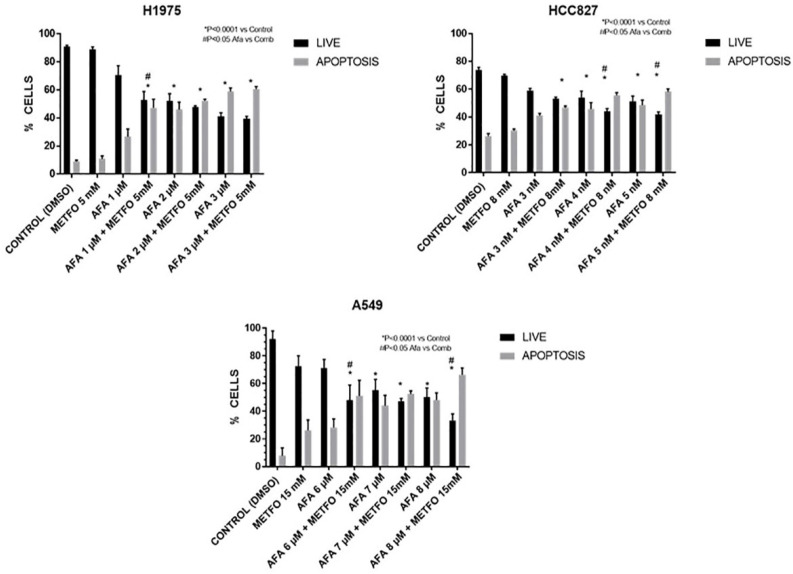
Apoptosis induction of afatinib plus metformin treatment in H1975, HCC827, and A549 cell lines. We observed similarities between the apoptosis test and cytotoxicity induction results. In total, 5000 events were analyzed in each assay. Cells were seeded and treated with the previously described scheme for 72 h and then analyzed with the apoptosis kit and flow cytometry. Bars represent the means of 3 independent experiments by triplicate. Statistical references are presented in each graph. * *p* < 0.0001 vs control, # *p* < Afa vs. Combo.

**Figure 3 pharmaceuticals-15-00381-f003:**
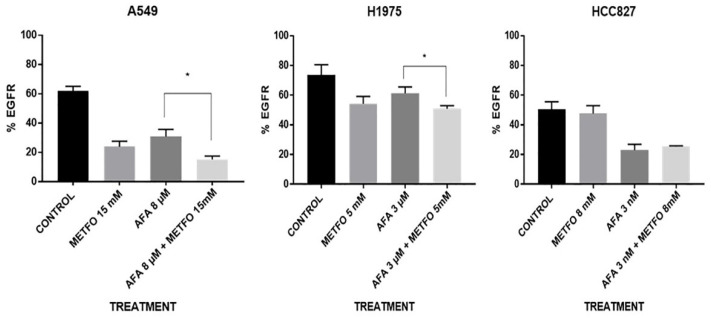
Membrane EGFR expression by metformin–afatinib. Cells were seeded and treated with respective schemes for 72 h, then, 5000 events were analyzed by flow cytometry with an EGFR-specific antibody. Bars represent the means of 3 independent experiments by triplicate. Statistical analysis was performed through one-way ANOVA. * *p* ≤ 0.05.

**Figure 4 pharmaceuticals-15-00381-f004:**
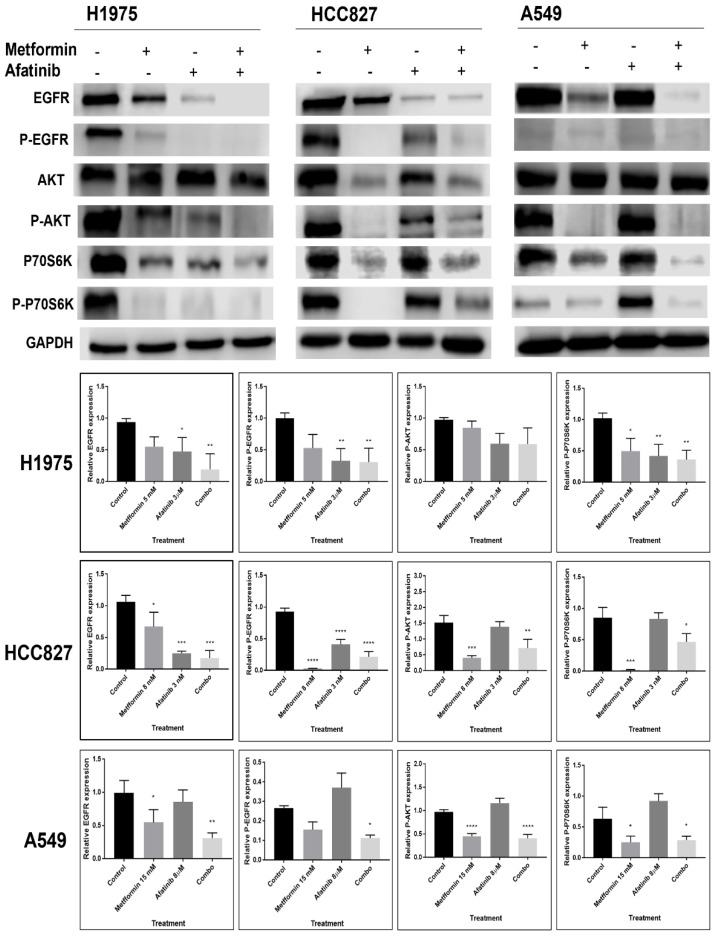
Effect of combination therapy metformin–afatinib on the EGFR signaling pathway. Cells were seeded and treated for 72 h with their respective metformin–afatinib concentrations. GAPDH was used as constitutive control, Western blot images were analyzed by image (NIH) and represented as bars. Images are representative of three independent experiments and results of area are presented as mean ± SD. Data were normalized regarding endogenous control and statistically analyzed by one-way ANOVA. * *p* ≤ 0.05, ** *p* ≤ 0.01, *** *p* ≤ 0.001, **** *p* ≤ 0.0001.

**Figure 5 pharmaceuticals-15-00381-f005:**
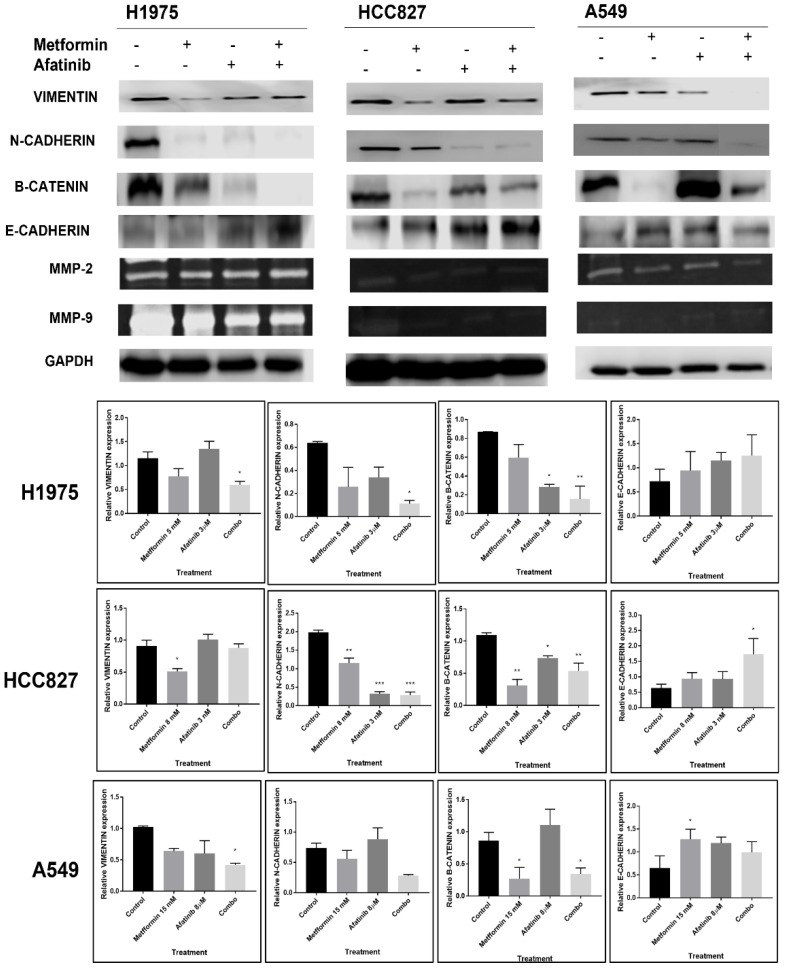
EMT biomarkers in NSCLC cell lines treated with metformin, afatinib, and the combination scheme. Cells were seeded and treated for 72 h with their respective metformin–afatinib concentrations. GAPDH was used as constitutive control, Western blot images were analyzed by image (NIH) and represented as bars. Images are representative of three independent experiments and results of area are presented as mean ± SD. For the zimogram assay, images are representative of two independent experiments. Data were normalized regarding endogenous control and statistically analyzed by one-way ANOVA. * *p* ≤ 0.05, ** *p* ≤ 0.01, *** *p* ≤ 0.001.

**Figure 6 pharmaceuticals-15-00381-f006:**
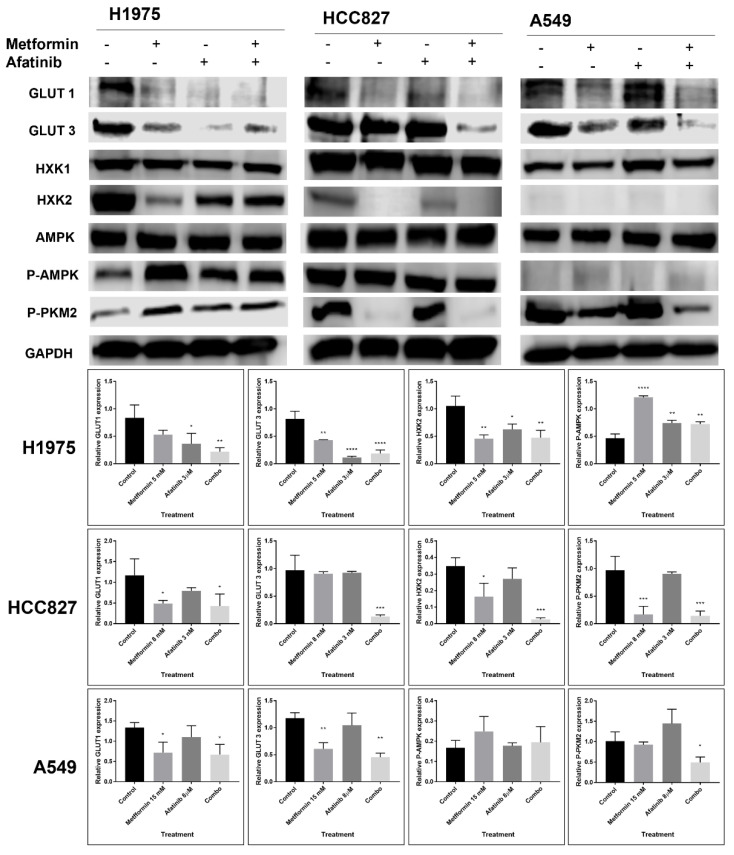
Effect of metformin–afatinib combined treatment on glycolytic enzymes and proteins. Cells were seeded and treated for 72 h with their respective metformin–afatinib concentrations. GAPDH was used as constitutive control, Western blot images were analyzed by image (NIH) and represented as bars. Images are representative of three independent experiments and results of area are presented as mean ± SD. Data were normalized regarding endogenous control and statistically analyzed by one-way ANOVA. * *p* ≤ 0.05, ** *p* ≤ 0.01, *** *p* ≤ 0.001, **** *p* ≤ 0.0001.

**Figure 7 pharmaceuticals-15-00381-f007:**
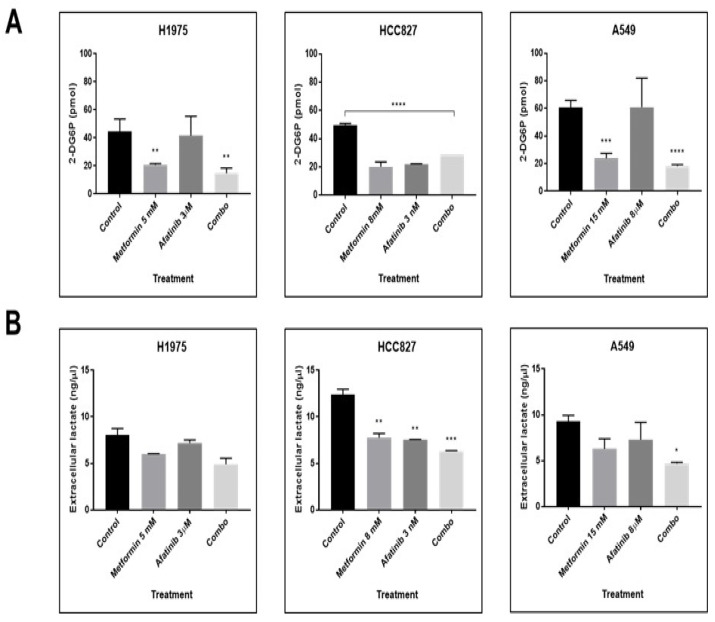
Cell glucose uptake and lactate secretion modifications. (**A**) For the glucose uptake assay, cells were seeded and metformin–afatinib treatment was administered in KRPH buffer over 3 h, then 2-DG6P was added and later its consumption was evaluated by ELISA. (**B**) Cells were seeded and later treated with metformin–afatinib for 3 h, levels of lactate present in the culture medium were measured by ELISA. Graphs represent the means of two independent experiments by duplicate. One-way ANOVA analysis was performed in order to determine statistical significance * *p* ≤ 0.05, ** *p* ≤ 0.01, *** *p* ≤ 0.001, **** *p* ≤ 0.0001.

**Figure 8 pharmaceuticals-15-00381-f008:**
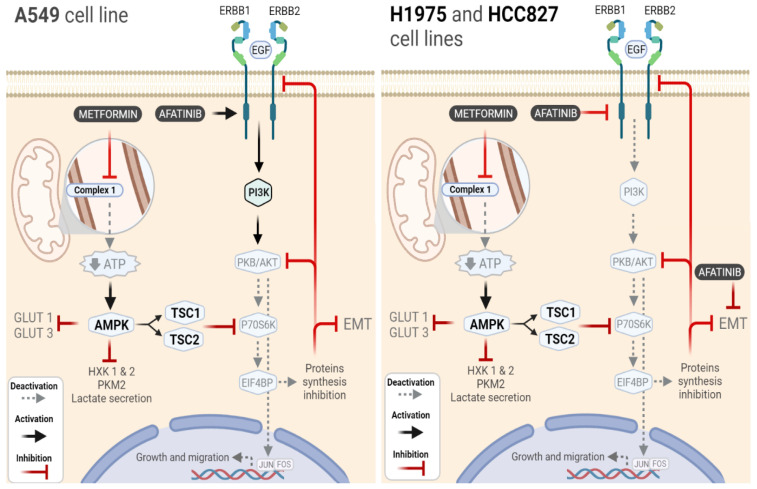
Mechanism of action of combined treatment afatinib–metformin. In the EGFR mutant LC cell lines (H1975 and HCC827), afatinib exerts its basal inhibitory effects over the EGFR pathway, decreasing both processes, glycolysis, and EMT transition. Furthermore, this inhibition can be exacerbated with the complementary effect of metformin through AMPK stimulation and subsequent P70S6K inhibition coupled with a decrease in protein synthesis. On the other hand, the A549 cell line (EGFR wild-type) showed stimulation of the EGFR pathway associated with afatinib treatment as a single drug, however, with complementary metformin treatment, the combination can counteract the pathway activation caused by afatinib, decreasing protein synthesis, glycolytic phenotype, and EMT; also, our results suggest a sensitization of this cell line to afatinib treatment when metformin is added, acting synergistically in cytotoxic induction.

**Table 1 pharmaceuticals-15-00381-t001:** Scheme of treatment concentrations for each cell line.

Cell Lines	Concentrations of Drugs	
	Metformin	Afatinib
A549	15 mM	6 µM (Combo 1)
		7 µM IC_50_ (Combo 2)
		8 µM (Combo 3)
H1975	5 mM	1 µM (Combo 1)
		2 µM IC_50_ (Combo 2)
		3 µM (Combo 3)
HCC827	8 mM	3 nM (Combo 1)
		4 nM IC_50_ (Combo 2)
		5 nM (Combo 3)

To evaluate modifications in metabolism, signaling pathways, and protein expression induced by treatment, we used the highest synergistic combos detected (depending on cytotoxicity) for each lung cancer cell line (combo 3 for A549 and H1975 cells and combo 1 for HCC827 cells).

## Data Availability

Data is contained within the article and Appendix A.

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
