# Peer review of "Metformin Enhances TKI-Afatinib Cytotoxic Effect, Causing Downregulation of Glycolysis, Epithelial–Mesenchymal Transition, and EGFR-Signaling Pathway Activation in Lung Cancer Cells"

_pharmaceuticals, 2022, doi:10.3390/ph15030381_

Round 1
Reviewer 1 Report
The authors examined synergic effects of metformin with EGFR-TKI (afatinib) in NSCLC. Although the data is interesting, there are some pointes to be further addressed.
- Novelty of this study is the use of second generation EGFR-TKI, afatinib. Please pmphasize this in the introduction section.
- It is interesting that metformin suppresses the expression of several proteins including EGFR, P70S6K. However, discussion about the possible mechanisms of this regulation is lacking in discussion section.
- In Figure 5, it is recommended to evaluate the expression of epithelial maker(s) as well as mesenchymal markers.
- Please include appropriate references (lines 257-262 and lines 267-268).
Author Response
Point 1. Novelty of this study is the use of second generation EGFR-TKI, afatinib. Please pmphasize this in the introduction section.
Response 1. We take your suggestion into account by adding information corresponding to second-generation TKIs and emphasizing its use and the possible mechanisms it modifies.
Point 2. It is interesting that metformin suppresses the expression of several proteins including EGFR, P70S6K. However, discussion about the possible mechanisms of this regulation is lacking in the discussion section.
Response 2. Information about this issue was added in some parts of the discussion section. We discussed the mechanism that involves AMPK, P70S6K, protein synthesis, and its repercussions on cell proliferation
Point 3. In Figure 5, it is recommended to evaluate the expression of epithelial maker(s) as well as mesenchymal markers.
Response 3. Thank you for this observation, it helped us to reinforce what was found with our mesenchymal markers. We performed experiments to evaluate the epithelial marker E-cadherin, finding a trend to increase in all treatments. This effect was more evident in Hcc827 with the combo, and in A549 when was treated with metformin. figure 5.
Point 4. Please include appropriate references (lines 257-262 and lines 267-268).
Response 4. Thank you for your observation, when making corrections, the references were moved in the text, we already placed them in the correct order.
Additionally, we take into account your suggestions of the general paper, and we made modifications to improve it.
Reviewer 2 Report
Review
This study sought to investigate the effects of a single agent vs. combination therapy with metformin and afatinib on three adenocarcinoma lung cancer cell lines, A549, H1975, and HCC827 cell lines, with different EGFR mutation status.
The most interesting results are seen in the reduction in expression of EGFR in metformin and afatinib treated cells, and also in N-Cadherin. The effect on GLUT1 and GLUT3 expression is also interesting as it indicates a modulation of transporters.
Altogether the data is sound.
Major points
The Discussion should be reduced as it is too long.
One major limitation is the lack of clinical data. In this context, some references should be discussed (e.g. Han, R.,et al. (2021). Concurrent use of metformin enhances the efficacy of EGFR-TKIs in patients with advanced EGFR-mutant non-small cell lung cancer—an option for overcoming EGFR-TKI resistance. Translational Lung Cancer Research, 10(3), 1277.) to enrich the discussion. These authors evaluated the efficacy of concurrent use of metformin with EGFR-TKIs, and assessed whether the addition of metformin may improve clinical outcomes and delay the occurrence of EGFR-TKI resistance. Also the effect of metformin in NSCLC Type 2 diabetes mellitus (T2DM) patients could enrich the discussion (Han, R., Jia, Y., Li, X., Zhao, C., Zhao, S., Liu, S., ... & Zhou, C. (2021). P76. 07 Metformin Enhances the Efficacy of EGFR-TKIs in Advanced Non-Small Cell Lung Cancer Patients With Type 2 Diabetes Mellitus. Journal of Thoracic Oncology, 16(3), S588).
Author Response
Point 1. The Discussion should be reduced as it is too long.
Response 1. We took your suggestion into account, we modified the entire discussion trying to focus on the relevant points of our investigation, and, omitting information that was repetitive since it was found in previous sections of the paper.
Point 2. One major limitation is the lack of clinical data. In this context, some references should be discussed (e.g. Han, R.,et al. (2021). Concurrent use of metformin enhances the efficacy of EGFR-TKIs in patients with advanced EGFR-mutant non-small cell lung cancer—an option for overcoming EGFR-TKI resistance. Translational Lung Cancer Research, 10(3), 1277.) to enrich the discussion. These authors evaluated the efficacy of concurrent use of metformin with EGFR-TKIs, and assessed whether the addition of metformin may improve clinical outcomes and delay the occurrence of EGFR-TKI resistance. Also the effect of metformin in NSCLC Type 2 diabetes mellitus (T2DM) patients could enrich the discussion (Han, R., Jia, Y., Li, X., Zhao, C., Zhao, S., Liu, S., ... & Zhou, C. (2021). P76. 07 Metformin Enhances the Efficacy of EGFR-TKIs in Advanced Non-Small Cell Lung Cancer Patients With Type 2 Diabetes Mellitus. Journal of Thoracic Oncology, 16(3), S588).
Response 2. Thank you for your bibliography recommendations, we add them in the clinical information of the discussion, to reinforce the previous clinical studies that we cited in the text.
Additionally, we improved the introduction section with some information and a better explanation of the mechanisms that we evaluated.
Round 2
Reviewer 1 Report
The manuscript is well revised and suitable for publication.
Reviewer 2 Report
The manuscript has been improved substantially and the data is interesting and valid, therefore acceptable.